# Operating Room Fomites as Potential Sources for Microbial Transmission in Burns Theatres

**Mariam Rela** [1,2,*]**, Sophia Opel** [1]**, Sarah Williams** [1]**, Declan P. Collins** [1,3]**, Kevin Martin** [4]**, Nabeela Mughal** [1,5,6] **and Luke S. P. Moore** [1,5,6]

1   Department of Plastic Surgery and Burns, Chelsea and Westminster Hospital NHS Foundation Trust, London SW10 9NH, UK; sophiaopel@gmail.com (S.O.); sawilliams@doctors.org.uk (S.W.); Declan.collins@chelwest.nhs.uk (D.P.C.); nabeela.mughal@nhs.net (N.M.); Luke.moore@nhs.net (L.S.P.M.)
2   Department of Plastic Surgery, University Hospitals Plymouth NHS Trust, Plymouth PL6 8DH, UK
3   Department of Surgery and Cancer, Imperial College London, London SW7 2AZ, UK
4   Department of Global Health and Infection, Brighton and Sussex Medical School, University of Sussex, Brighton BN1 9PX, UK; kevinmartin@doctors.org.uk
5   North West London Pathology, London W6 8RF, UK
6   NIHR Health Protection Research Unit in Healthcare Associated Infections and Antimicrobial Resistance, Imperial College London, Hammersmith Campus, London W12 0NN, UK
*   Correspondence: Mariam.rela@nhs.net; Tel.: +02-033-158-273

**Abstract:** Background: Burn patients are susceptible to healthcare-associated infections. Contaminated surfaces play a role in microbial transmission. This study aimed to quantify the degree of contamination of burns theatre fomites during routine clinical use. Methods: The Patslide Patient Transfer Board (PAT slide) and operating table were investigated using two methods—bacterial swabs to culture viable organisms and adenosine triphosphate (ATP) swabs to measure biological material. Both items were sampled four times a day: before the first case, immediately after a case, immediately before the next case after cleaning and after the terminal clean. Results: Among 82 bacterial samples, four organisms were isolated, including *Staphylococcus aureus*, *Enterobacter cloacae* (*E. cloacae*) x2 and *Pseudomonas aeruginosa* (*P. aeruginosa*), all from the PAT slide. The *E. cloacae* persisted after cleaning. In 9/82 swabs, the ATP count was >10 relative light units (RLU). In all cases where an organism was identified, the ATP count was >10 RLU. Hence the sensitivity and specificity of ATP > 10 RLU in detecting an organism were 100% and 94% respectively. Conclusions: Within burns theatres, there are instances of bacterial contamination on surfaces that persist despite cleaning. ATP luminometers as a point-of-care device may have a role in determining the cleanliness of surfaces, potentially minimizing onwards-bacterial transmission.

**Keywords:** burns surgery; infection control; environmental contamination; healthcare-associated infections

## 1. Introduction

Burn patients are particularly susceptible to healthcare-associated infections. This is primarily due to a loss of skin integrity which acts as a barrier against micro-organisms [1]. Moreover, burns are associated with a dysregulation of the innate and adaptive immune responses, further predisposing these patients to infection [2]. There is an increased incidence of multi-drug resistant (MDR) organisms with longer hospital stays [3], which may include meticillin-resistant *Staphylococcus aureus* (MRSA) and multidrug resistant strains of *P. aeruginosa* and *Acinetobacter baumannii* [1,3]. Moreover, as a result of the impaired immune responses in burn patients, viruses, for example herpes simplex and cytomegalovirus, may also invade burn wounds, contributing to infection [4]. Infections in burn patients are a significant cause of morbidity and mortality [3]. Factors that increase

the risk of this include delayed burn care, prolonged open wounds, a higher total body surface area (TBSA) (>30%) and significant full thickness burns [5].

There is increasing evidence linking the acquisition of healthcare-associated infections with contaminated surfaces [5,6]. The mode of transmission can be related to direct contact with contaminated surfaces, with bodily fluids such as blood, or indirectly from the hands of healthcare workers [5]. Not only are burn patients unique in their acquisition of infections, with the risk increasing the greater the TBSA of the burn wound, but in addition, burn patients disperse a vast quantity of organisms into the environment, with greater volumes being shed in patients with a higher TBSA [5]. Some nosocomial pathogens can persist on inanimate surfaces for long periods [7,8]. Some persist despite routine cleaning and disinfection procedures [9].

A recent study from the Birmingham (UK) burns unit in 2015 highlighted the link between patient colonisation and environmental contamination. The burns shock room in which an internationally transferred patient colonised with carbapenemase-producing organisms (CPOs) was cared for, showed evidence of multiple CPOs on environmental sampling despite a routine terminal clean [10]. Secondly, authors reporting an outbreak in the Mersey (UK) burns unit in which 9 patients contracted an MDR *Pseudomonas*. spp. following the international transfer of a colonised patient in 2015, hypothesised that the burns service environment was a likely source of transfer of the organism [11].

To explore the degree of microbial contamination of burns theatre equipment during routine clinical use, we undertook a single-centre, prospective, observational study. We ascertained the utility of routine cleaning procedures in theatre to reduce biological and microbial contamination and hence potential infection transmission to vulnerable burn patients.

## 2. Materials and Methods

### 2.1. Setting and Study Design

The Chelsea and Westminster Hospital Burns Unit is a specialist service for children and adults. The adult unit comprises two intensive care beds, two high-dependency beds, nine ward beds, one theatre and a busy clinic service. Two operating theatre fomites were identified which commonly come into direct contact with the patient: the PAT slide and operating table. These pieces of equipment are routinely cleaned between cases and at the end of the day with *Chlor-clean* (0.1% (1000 ppm) chlorine solution) and a j cloth according to established standard operating procedure and in line with manufacturer guidelines. The *Chlor-clean* is prepared by one (6 g) tablet being dissolved in 1 L cold water and this provides a solution that both cleans and disinfects surfaces. *Chlor-clean* has been shown to be effective against viruses, fungi and infection-causing bacteria such as *Escherichia coli (E.Coli)*, *Pseudomonas*, *Klebsiella*, *Staphylococcus*, *Enterococcus*, *Acinetobacter* and *Clostridium difficile* [12,13].

### 2.2. Sampling Protocol

Sampling was performed on 11 non-consecutive weekdays from March to May 2019. Two modes of detection of bacterial contamination were utilised, including (i) microbiological swabs to culture viable organisms and (ii) adenosine triphosphate (ATP) swabs (*Hygiena luminometer and ATP Ultrasnap surface swabs, Complete Safety Supplies*) as a semi-quantitative measure of residual biological material. The process for the collection of swabs is detailed in Figure 1.

Swabs were taken by burns registrars from these two sites four times a day: once in the morning before starting the operating list, once immediately after a case, once before the next case after cleaning and lastly at the end of the day after the terminal clean. The entire PAT slide could not be sampled so a cross-shaped pattern from corner to corner on

the side that came into contact with the patient was swabbed. The lower part of the operating table which came into contact with lines and catheters attached to the patient was also swabbed (Figure 1).

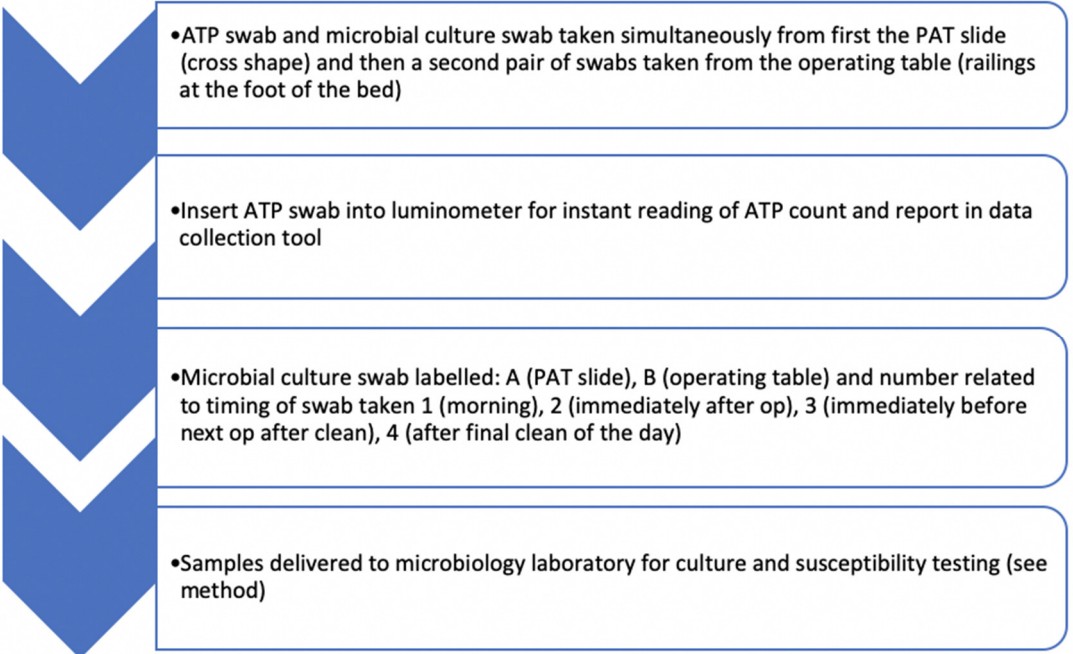

**Figure 1.** Sampling protocol to investigate burns operating theatre fomites as potential sources for microbial transmission, March to May 2019, London.

### 2.3. Determining Biological Contamination of Operating Theatre Fomites

Adenosine triphosphate (ATP) is present in all living cells and can act as an indicator for microbial contamination [14]. It has previously been used in the hospital setting to determine relative contamination of various healthcare environment items [15]. The assay contains luciferin-luciferase which generates light when it comes into contact with ATP, which is quantified by the luminometer into relative light units (RLU) [16,17]. The manufacturer guidance for the *Hygiena* ATP luminometer used specified a cut-off of >10 RLU as a marker of biological contamination. Results were available at point-of-care and recorded directly.

### 2.4. Determining Microbiological Contamination of Operating Theatre Fomites

Microbiological swabs were set up for culture in a method analogous to the UK standards for microbiological investigation for swabs from skin and superficial soft tissue infections [18]. Standard microbiological swabs were plated onto non-selective blood agar and MacConkey agar. Plates were incubated for 16–24 h at 37 °C in an aerobic atmosphere. Organisms grown were identified using matrix-assisted laser desorption/ionization time of flight (MALDI-ToF; biotyper®, Bruker Daltonik GmbH, Bremen, Germany). Antimicrobial susceptibility was discerned using disc diffusion against European Committee on Antimicrobial Susceptibility Testing (EUCAST; v9.0) [19].

### 2.5. Statistical Analysis

Data analysis was performed using STATA version 15.0 (StataCorp, College Station, TX, USA). The performance characteristics (sensitivity and specificity) of ATP detection as a measure of biological contamination were calculated using isolation of a bacterial isolate as the gold standard test. The proportion of swabs with ATP > 10 RLU from the

PAT slide and operating table were compared using Fisher's exact test. Pre-and post-clean ATP results were compared using the Wilcoxon-signed rank test.

### 2.6. Study Approval

This study was registered as a service evaluation with the Chelsea and Westminster NHS Foundation Trust Audit and Governance Office (registration number: pcd789).

### 3. Results

There were 41 distinct time intervals of sampling the operating theatre fomites; in 3 instances there was only one case in theatre, hence swabs could not be collected after cleaning before the next case. There were 82 results each for ATP luminometer readings and microbial culture, the results are displayed in Table 1.

**Table 1.** Adenosine triphosphate (ATP) and microbial culture results from investigation of burns operating theatre fomites (The Patslide Patient Transfer Board (PAT) slide and operating table) as potential sources for microbial transmission, March to May 2019, London.

| Day | Operating Theatre Fomite Sampled | Start of Day ATP Count (RLU) | Microbial Culture Results | After Case Before Cleaning ATP Count (RLU) | Microbial Culture Results | Before Case after Cleaning ATP Count (RLU) | Microbial Culture Results | After Final Clean ATP Count (RLU) | Microbial Culture Results |
|---|---|---|---|---|---|---|---|---|---|
| 1 | PAT slide | 0 | NSG | 0 | NSG | 2 | NSG | 0 | NSG |
| | Operating table | 1 | NSG | 1 | NSG | 2 | NSG | 0 | NSG |
| 2 | PAT slide | 0 | NSG | 1 | NSG | 0 | NSG | 0 | NSG |
| | Operating table | 0 | NSG | 1 | NSG | 1 | NSG | 2 | NSG |
| 3 | PAT slide | 0 | NSG | 16 * | MSSA * | N/A | N/A | 0 | NSG |
| | Operating table | 6 | NSG | 0 | NSG | N/A | N/A | 0 | NSG |
| 4 | PAT slide | 1 | NSG | 1 | NSG | 0 | NSG | 33 * | NSG |
| | Operating table | 0 | NSG | 1 | NSG | 1 | NSG | 1 | NSG |
| 5 | PAT slide | 0 | NSG | 1 | NSG | N/A | N/A | 0 | NSG |
| | Operating table | 0 | NSG | 0 | NSG | N/A | N/A | 0 | NSG |
| 6 | PAT slide | 0 | NSG | 0 | NSG | 0 | NSG | 0 | NSG |
| | Operating table | 0 | NSG | 0 | NSG | 0 | NSG | 0 | NSG |
| 7 | PAT slide | 3 | NSG | 14 * | NSG | 1 | NSG | 0 | NSG |
| | Operating table | 0 | NSG | 2 | NSG | 1 | NSG | 1 | NSG |
| 8 | PAT slide | 3 | NSG | 24 * | Enterobacter cloacae * | N/A | N/A | 36 * | Enterobacter cloacae * |
| | Operating table | 0 | NSG | 2 | NSG | N/A | N/A | 0 | NSG |
| 9 | PAT slide | 0 | NSG | 2 | NSG | 0 | NSG | 1 | NSG |
| | Operating table | 0 | NSG | 0 | NSG | 0 | NSG | 1 | NSG |
| 10 | PAT slide | 1 | NSG | 1 | NSG | 0 | NSG | 0 | NSG |
| | Operating table | 1 | NSG | 0 | NSG | 1 | NSG | 0 | NSG |
| 11 | PAT Slide | 0 | NSG | 149 * | Pseudomonas aeruginosa * | 102 * | NSG | 0 | NSG |
| | Operating table | 0 | NSG | 61 * | NSG | 161 * | NSG | 2 | NSG |

Legend: N/A = Only 1 case operated on that day, NSG = No significant growth, * = ATP count >10 relative light units (RLU) or organism culture.

### 3.1. Determining Biological Contamination of Operating Theatre Fomites through ATP Detection

In 9/82 swabs, the ATP count was >10 RLU (Figure 2). Five of these swabs were taken immediately after a case, and in these five cases, there was no significant reduction in ATP

count after cleaning (pre-clean median 24 RLU (IQR 16–61) vs. post-clean median 36 RLU (IQR 1–102); *p* = 0.69). There was no significant difference between the PAT slide and the operating table regarding the proportions with high (>10 RLU) ATP readings (17.1% vs. 4.9%; *p* = 0.155).

### 3.2. Determining Microbiological Contamination of Operating Theatre Fomites Through Microbiological Culture

Among 82 bacterial samples taken, four organisms were isolated, including (meticillin susceptible) *Staphylococcus aureus (MSSA)*, *E. cloacae* (x2) and *P. aeruginosa,* all from the PAT slide (Figure 2). Three swabs where organisms were isolated were taken immediately after a case. In the episode where an *E. cloacae* was isolated immediately after a case, it persisted after cleaning. In all cases where a bacterial isolate was cultured, the ATP count was >10 RLU. The sensitivity and specificity of an ATP >10 RLU as a measure of significant microbial culture were 100% and 94% respectively.

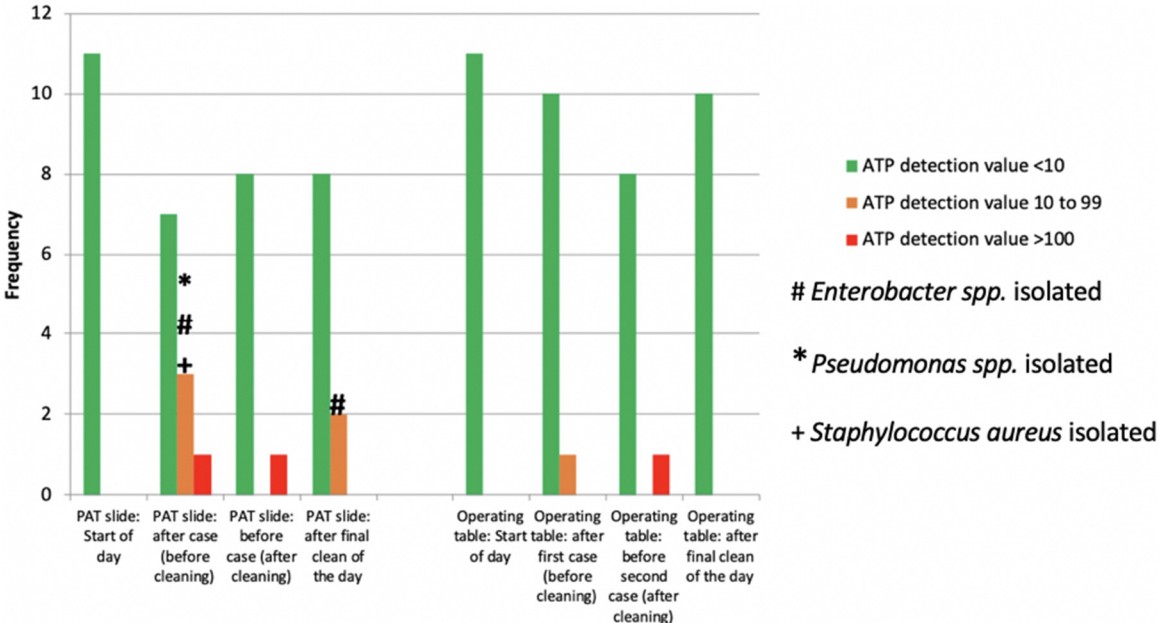

**Figure 2.** Adenosine triphosphate and microbial culture results from investigation of burns operating theatre fomites (PAT slide and operating table) as potential sources for microbial transmission, March to May 2019, London.

### 3.3. Clinical Correlation Between Patient Colonisation and Microbial Contamination

There were three days during the study in which bacteria were isolated from the fomites. Over these three days combined, four patients were operated on in the burns theatre. None of these patients developed signs of clinical infection but were all colonised with bacteria pre- and intra-operatively. On the day an MSSA was isolated, the one patient in theatre was colonised with *P. aeruginosa* pre- and intra-operatively and developed no new infections post-operatively. The day that the *E. cloacae* was isolated immediately after the case and after cleaning, the one patient was colonised with an AmpC producing *E. Cloacae* pre- and intra-operatively as well as with other organisms including an Extended Spectrum Beta-Lactamase *E. Coli*, *Enterococcus spp.* and *Bacillus spp.* There were no changes in the patient's colonisation status post-operatively. On the day that *P. aeruginosa* was isolated, the first patient was colonised with this pre-operatively. The second patient of the day was colonised with *Staphylococcus Aureus, Streptococcus spp.* and *Bacillus spp.* pre-operatively but post-operatively developed nosocomial colonisation with *Pseudomonas putida*.

## 4. Discussion

We find overall low levels of biological contamination and significant bacterial isolates on two burns operating theatre fomites which have significant contact with patients. Furthermore, we find that point-of-care ATP detection has respectable test performance characteristics in predicting the likelihood of subsequent culturable bacteria and as such could conceivably have a role in real-time audit of burns operating theatres.

Regarding the microbial culture findings, no organisms were isolated at the start of the day and all instances of initial detection of microbial contamination occurred immediately after a case, before the theatre had been cleaned. This is in keeping with the theory that burn patients shed organisms into their surroundings [5,9], particularly given that with the swabs taken immediately after a case that isolated *E. cloacae* and *P. aeruginosa,* the patients were colonised with these organisms pre-operatively. The finding of *E. cloacae* on the PAT slide persisting following the terminal clean raises the possibility that organisms have the potential to be transferred between patients and contribute to healthcare-associated infections. Particularly poignant is that on the day that *P. aeruginosa* was isolated after the first case, the second patient developed a nosocomial colonisation with a different *Pseudomonas* spp. which may have been transmitted intra-operatively from environmental contamination. Whilst possible, it does not automatically follow that surface microbial contamination, when in contact with a patient, may later manifest as clinical infection.

The burns theatre environment would be expected to have high hygiene standards and as such the ATP values measured were very low (77% of all RLU values were 0 or 1). A review assessing the effectiveness of ATP bioluminescence in assessing hygiene in hospital settings explains that studies have used a variety of RLU benchmark values to determine whether a surface is clean, ranging from 45 to 1000, the mostly commonly used of which is 250 RLU, for a number of luminometer brands [20]. A 2017 study using ATP bioluminescence to assess cleanliness in orthopaedic theatres had mean RLU values of 1054 for the preparation table and 2539 for their operating table headboard [21], values much higher than those in our study. It has been suggested that differing thresholds should be used for differing environments and surfaces, depending on the patient cohort, level of contact of surface with the patient and surface area/shape of surface [22]. One study found that flat surfaces were more likely to "pass" ATP cleanliness thresholds than irregularly shaped surfaces [22]. The "pass" limit of the *Hygiena* luminometer used in this study was 10, which was much lower than other benchmarks previously quoted. This limit correlated with our findings that above this, the specificity of an organism being isolated was 94%.

ATP bioluminescence has a number of advantages when used as a tool to monitor adequacy of cleaning in the hospital setting: it is rapid, real-time, quantitative and easy to use [17]. ATP bioluminescence allows temporal and spatial quantification of cleanliness, thus specific areas can be targeted with more intensive cleaning practices to reduce bacterial load. This was shown to be successful in a *P. aeruginosa* outbreak in a Swiss burns unit whereby targeted disinfection procedures in the hydrotherapy room, which was found to be a likely reservoir of pathogens, contributed to the control of the outbreak [23]. Moreover, educational interventions directed towards cleaning staff has been shown to significantly improve surface ATP counts in a hospital setting in Brazil [14]. In addition to rigorous disinfection, the optimal way to reduce the incidence of healthcare-associated infections is a combination of good hand hygiene, effective patient screening and isolation, optimising patient selection for surgical intervention and responsible antimicrobial stewardship [3,24].

The primary limitation of the study is its sample size, limited by the pragmatic nature of the study. This likely contributed to the non-significant results between pre- and post-cleaning for the instances of high ATP count. Furthermore, there are a number of reasons for hesitation in widespread adoption of ATP bioluminescence as a correlation for microbial burden. Firstly, the luminometers detect ATP in all cells, these might be patient cells

from skin or blood which commonly contaminate surfaces in theatre [17]. Furthermore, they cannot determine whether the bacteria detected were viable [22]. Moreover, the impact of detergents and disinfectants on the bioluminescence assay needs further analysis [17], and may reduce the validity of the scores. The Hawthorne effect can be explained as a change of behaviour in response to being observed or assessed [25], this may have played a part with the theatre staff's cleaning behaviour during the period of data collection [22,26].

## 5. Conclusions

Bacterial contamination on surfaces in a burns theatre is infrequent, but can occur despite routine infection control practices. This raises the possibility of organisms being transferred between patients and contributing to patient morbidity and mortality. ATP luminometers as a point-of-care device may have a role in determining the cleanliness of surfaces in high-risk areas such as operating theatres. This could allow for targeted cleaning interventions or microbial environmental monitoring where RLU values are high in order to reduce the risk of onwards transmission of bacteria.

**Author Contributions:** Designed the study methodology, M.R., L.S.P.M., N.M. and D.P.C.; Collected the data, S.O. and S.W.; Collated the data, M.R. and L.S.P.M.; Analysed the data, M.R. and K.M. All authors reviewed the results and data analysis and contributed comments. M.R. drafted the initial manuscript with all authors contributing significantly to revising this for submission. All authors have read and agreed to the published version of the manuscript.

**Funding:** This study was funded by a grant from the CW + Charity.

**Acknowledgments:** L.S.P.M. acknowledges support from The National Institute of Health Research (NIHR) Imperial Biomedical Research Centre (BRC) and The National Institute for Health Research Health Protection Research Unit (HPRU) in Healthcare Associated Infection and Antimicrobial Resistance at Imperial College London in partnership with Public Health England. The views expressed in this publication are those of the authors and not necessarily those of the NHS, The National Institute for Health Research, or the UK Department of Health.

**Conflicts of Interest:** L.S.P.M. has consulted for bioMerieux (2013–2020), DNAelectronics (2015–2018), Dairy Crest (2017–2018), Umovis Lab (2020), and Pfizer (2018–2020) received speaker fees from Profile Pharma (2018), received research grants from the National Institute for Health Research (2013–2020), CW + Charity (2018–2020), and Leo Pharma (2016), and received educational support from Eumedica (2016–2018). NM has received speaker fees from Beyer (2016) and Pfizer (2019) and received educational support from Eumedica (2016) and Baxter (2017). All other authors have no conflicts of interest to declare.

**Ethics Approval and Consent:** Ethics approval and consent were not needed for this environmental study.

**Data Availability:** All data is contained within the article.

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
