# Peer review of "Operating Room Fomites as Potential Sources for Microbial Transmission in Burns Theatres"

_2673-1991, doi:10.3390/ebj2010001_

Round 1

Reviewer 1 Report

Remarks:

Abstract, main text

  • Adding full description for the PAT slide (Patslide Patient Transfer Board) once, would be advisable.

Keywords: Burns surgery; plastic surgery; microbiology; infection control; decontamination

  • Keywords do not reflect the studies design and results; I would recommend changing the keywords.

 Materials and Methods section 2.1. Setting and study design: ‘Chlor clean’

  •  I would suggest clarifying in parenthesis: 0.1% (1,000ppm) chlorine solution

 Statistical analysis, ‘ATP cut-off of > 10 RLU was chosen a priori, as per manufacturer guidance.’

  • I suggest referring to the ATP cut-off point in the section 2.3. Determining biological contamination of operating theatre fomites.

Abstract, Results: Pseudomonas spp

  • I would suggest clarifying which pathogen of the Pseudomonadaceae family was isolated.

Discussion, last paragraph, ‘Lastly, theatre staff may have changed their behaviour during the period of data collection leading to a potential Hawthorne effect [19][20].’

  • Providing brief explanation of the Hawthorne effect would be advisable.

  • I suggest moving the 4th paragraph with the descriptions of the study limitations at the end of the Discussion section.

  • It would be helpful to provide information on the clinical relevance of microbial contamination of the operating theatre in all samples with positive culture. It would be interesting to know if microbial contamination of surfaces was linked to clinical cases.

Author Response

Abstract, main text

  • Adding full description for the PAT slide(Patslide Patient Transfer Board) once, would be advisable.
    • ‘Patslide patient transfer board’ added to line 18-19 in Abstract

Keywords: Burns surgery; plastic surgery; microbiology; infection control; decontamination

  • Keywords do not reflect the studies design and results; I would recommend changing the keywords.
    • Keywords on lines 31-32 changed to: Burns surgery; infection control; environmental contamination; healthcare-associated infections

 Materials and Methods section 2.1. Setting and study design: ‘Chlor clean’

  • I would suggest clarifying in parenthesis: 0.1% (1,000ppm) chlorine solution
    • 0.1% (1,000ppm) chlorine solution and j cloth added to line 73 in Materials and Methods

 Statistical analysis, ‘ATP cut-off of > 10 RLU was chosen a priori, as per manufacturer guidance.’

  • I suggest referring to the ATP cut-off point in the section 2.3. Determining biological contamination of operating theatre fomites.
    • ‘The manufacturer guidance for the Hygiena ATP luminometer used specified a cut-off of >10 RLU as a marker of biological contamination’ added to section 2.3 and , ‘ATP cut-off of > 10 RLU was chosen a priori, as per manufacturer guidance.’ removed from section 2.5.

Abstract, Results: Pseudomonas spp

  • I would suggest clarifying which pathogen of the Pseudomonadaceae family was isolated.
    • Expanded to Pseudomonas aeruginosa on line 24 of abstract with abbreviation P. aeruginosa which is used in the rest of the text.

Discussion, last paragraph, ‘Lastly, theatre staff may have changed their behaviour during the period of data collection leading to a potential Hawthorne effect [19][20].’

  • Providing brief explanation of the Hawthorne effect would be advisable.
    • Explanation added to lines 210-211 with reference ‘The Hawthorne effect can be explained as a change of behaviour in response to being observed or assessed’ in Discussion

  • I suggest moving the 4thparagraph with the descriptions of the study limitations at the end of the Discussion section.
    • Moved paragraph on limitations to the end of the Discussion section

  • It would be helpful to provide information on the clinical relevance of microbial contamination of the operating theatre in all samples with positive culture. It would be interesting to know if microbial contamination of surfaces was linked to clinical cases.

    • Section 3.3 added to results with new clinical information in which the cases in theatre on the days where environmental microbial contamination was found were investigated for clinical infection and wound colonisation by reviewing patient notes and microbiology results.
    • Relevance of this new information added to discussion on lines 169-170 ‘Particularly given that with the swabs taken immediately after a case that isolated cloacae and P. aeruginosa, the patients were colonised with these organisms pre-operatively.; and 173-175 ‘Particularly poignant is that on the day that P. aeruginosa was isolated after the first case, the second patient developed a nosocomial colonisation with a different Pseudomonas spp. which may have been transmitted intra-operatively from environmental contamination.’

Reviewer 2 Report

Dear authors,

generally, the work is organized properly, however, several minor/major corrections should be done and these are the following:

  • The authors should clearly indicate the chemicals that were used for cleaning and/or disinfection of the fomites. Please add.
  • In the abstract - the abbreviated name of Enterobacter cloacae should be mentioned in the bracket just after the full name of this bacterium. Then the abbreviation can be used thereafter in the whole manuscript. Please correct.
  • Please correct the phrase 'Burns patients' into 'burn patients' in the whole text. 
  • Lin 39 - The authors have mentioned the bacterial infections that are obviously most common, however, in terms of burn patients, also viral infections should be mention since they also contribute to the increase of mortality and morbidity rates of burn patients. See Baj et al. 2020.
  • Line 43 - the hands are not the only source of a possible infection, also the bodily secretions like blood are crucial in terms of transmission. Please add.
  • Line 44 - the authors should mention the term TBSA - total body surface area.
  • In the introduction section, the authors could mention the division of the burns according to their degree. This might provide the readers with an outlook on the fact that for instance burns of the third degree might be associated with a greater risk of infection.
  • In the setting and study design, only the setting is described without any materials used in this study for instance the kits for a swab collection. The authors did not mention the cleaning chemicals and their spectrum of bacterial killing which is crucial to add here. Please correct.

Author Response

  • The authors should clearly indicate the chemicals that were used for cleaning and/or disinfection of the fomites. Please add.
    • Further information about Chlor-clean, how it was used and the organisms it is effective against added to lines 72-78 of Methods with references.
  • In the abstract - the abbreviated name of Enterobacter cloacae should be mentioned in the bracket just after the full name of this bacterium. Then the abbreviation can be used thereafter in the whole manuscript. Please correct.
    • Abbreviation added to line 24 in Abstract, Enterobacter cloacae shortened to E. cloacae in line 138 of Results
  • Please correct the phrase 'Burns patients' into 'burn patients' in the whole text. 
    • Corrected throughout text in lines 16, 35, 41, 42, 49, 50, 65, 168
  • Lin 39 - The authors have mentioned the bacterial infections that are obviously most common, however, in terms of burn patients, also viral infections should be mention since they also contribute to the increase of mortality and morbidity rates of burn patients. See Baj et al. 2020.
    • Baj et al reference included and lines 40-42 added in introduction mentioning viral infections of burn wounds in addition to bacterial.
  • Line 43 - the hands are not the only source of a possible infection, also the bodily secretions like blood are crucial in terms of transmission. Please add.
    • ‘with bodily fluids such as blood’ Added to line 48 of introduction
  • Line 44 - the authors should mention the term TBSA - total body surface area.
    • Reference made to burns of higher TBSA being more likely to disperse organisms into the environment and acquire infections in lines 50 and 51
    • Burns with higher TBSA associated with higher morbidity and mortality with infection on line 44.
  • In the introduction section, the authors could mention the division of the burns according to their degree. This might provide the readers with an outlook on the fact that for instance burns of the third degree might be associated with a greater risk of infection.
    • In line 45 of introduction, reference made to increased risk of morbidity and mortality associated with infection where there are significant full thickness burns.
  • In the setting and study design, only the setting is described without any materials used in this study for instance the kits for a swab collection. The authors did not mention the cleaning chemicals and their spectrum of bacterial killing which is crucial to add here. Please correct.
    • Further information on Chlor-clean and how it is used added to Methods Section 2.1 lines 73-78 regarding Chlor-Clean.
    • Hygiena luminometer and ATP surface swabs used clarified on line 82.

Round 2

Reviewer 2 Report

Dear authors

thank You for correcting the manuscript according to my comments. 
i have No further suggestions.